# Sex, Age, and COVID-19 Vaccine Characteristics Associated with Adverse Events After Vaccination and Severity: A Retrospective Analysis

**DOI:** 10.3390/idr17050108

**Published:** 2025-09-03

**Authors:** Edgar P. Rodríguez-Vidales, Jesús M. Santos-Flores, Mara I. Garza-Rodríguez, Ana M. Salinas-Martínez, Alejandra G. Martínez-Pérez, Roberto Montes de Oca-Luna, Alma R. Marroquín-Escamilla

**Affiliations:** 1Health Services of the State of Nuevo Leon, Ministry of Health, Monterrey 64000, NL, Mexico; edgar.rodriguez@saludnl.gob.mx (E.P.R.-V.); biol.maragarza@gmail.com (M.I.G.-R.); roberto.montesdeoca@saludnl.gob.mx (R.M.d.O.-L.); 2General Hospital of Sabinas Hidalgo, Ministry of Health, Sabinas Hidalgo 65270, NL, Mexico; jesus.santos@saludnl.gob.mx; 3Epidemiologic and Health Services Research Unit, Mexican Institute of Social Security, Monterrey 64360, NL, Mexico; amsalinasmartinez@gmail.com; 4Department of Histology, School of Medicine, Autonomous University of Nuevo Leon, Monterrey 64460, NL, Mexico; alejandra.martinezpr@uanl.edu.mx

**Keywords:** COVID-19 vaccination, pharmacovigilance, epidemiology, vaccine safety

## Abstract

Background: Although the safety of COVID-19 vaccines has been demonstrated in clinical trials, real-world pharmacovigilance remains essential to detect rare or unexpected adverse events following immunization (AEFI). In Mexico, the national AEFI surveillance system is in place, yet there is limited analysis of state-level data. Objective: To characterize AEFI related to five COVID-19 vaccines and identify factors associated with AEFI type and seriousness in Nuevo León, Mexico. Methods: A retrospective analysis of the State of Nuevo León AEFI database was conducted, including all AEFI reports between December 2020 and June 2022 (n = 2213). Data included patient sex, age, vaccine type (Pfizer/BioNTech, AstraZeneca, Sinovac, Moderna, CanSino), number of doses (1 or ≥2), symptom categories, and AEFI seriousness. Symptoms were classified as local or systemic and grouped by organ systems. Descriptive analysis and binary multivariate logistic regression were used to examine associations between demographic and vaccine-related factors with AEFI type and severity. Odds ratios (OR) with 95% confidence intervals (CI) were estimated. Results: Most AEFI reports involved females aged 19–59 years and occurred after the first vaccine dose. The most frequently reported unexpected adverse events (UAEs) were mild to moderate, including injection-site reactions, headache, chills, fatigue, nausea, fever, dizziness, weakness, myalgia, and tachycardia. The Pfizer/BioNTech vaccine was associated with higher odds of arm pain and lower odds of hemorrhagic events. Receiving ≥2 doses increased the odds of arm pain and systemic symptoms. Less than 3% of AEFIs were classified as serious. Older adults (≥65 years) and second vaccine doses were associated with increased odds of a serious AEFI, while female sex and receiving the Pfizer/BioNTech vaccine were associated with reduced odds. Conclusions: In Nuevo León, most AEFIs related to COVID-19 vaccination were mild to moderate and resolved without complications. Serious AEFIs were uncommon, with older age and second doses associated with higher risk, and female sex and Pfizer/BioNTech vaccination associated with lower risk. These findings provide a local perspective on vaccine safety that complements national and international evidence.

## 1. Introduction

Since the emergency use authorization of COVID-19 vaccines in late 2020, global efforts focused on rapidly deploying vaccination campaigns to reduce transmission, morbidity, and mortality. In Mexico, the National Vaccination Strategy included several vaccine platforms such as mRNA (Pfizer/BioNTech, Moderna), inactivated virus (Sinovac, Sinopharm), non-replicating viral vectors (AstraZeneca, Sputnik V, CanSino), and protein subunit vaccines (Abdala). These were administered sequentially across prioritized populations, including healthcare workers, the elderly, and high-risk groups [1,2].

The safety of COVID-19 vaccines has been demonstrated in clinical trials, which have reported a low incidence of serious adverse events. Most reported reactions have been mild to moderate, including injection-site reactions and generalized symptoms such as headache, chills, fatigue, nausea, fever, dizziness, weakness, myalgia, and tachycardia [3,4,5]. Observational studies continue to identify adverse events, primarily through self-reported surveys [6,7,8,9] and epidemiological surveillance systems, both passive and active. An adverse event following immunization (AEFI) is defined as any untoward medical occurrence following vaccination that does not necessarily have a causal relationship with the vaccine itself [10]. Pharmacovigilance plays a crucial role in detecting rare or unexpected events that may not have been identified during pre-authorization clinical trials, especially under the expedited timelines followed during the COVID-19 pandemic. Rare adverse events (RAEs) are defined as adverse events that occur infrequently and are not commonly associated with vaccination. In contrast, unexpected adverse events (UAEs) refer to events not previously identified in clinical trials or established post-marketing surveillance at the time of vaccine administration [11]. Transparency and effective communication regarding the risks and benefits of vaccination are essential to maintaining public trust in immunization programs. Consequently, post-authorization monitoring is vital for accurately characterizing the safety profiles of vaccines.

In Mexico, the AEFI reporting system is coordinated by the General Directorate of Epidemiology and the National Center for Preventive Programs and Disease Control (CENSIA), with reports submitted through digital platforms and reviewed by expert committees [12,13]. Health professionals across all states are mandated to report suspected cases, which are then classified by severity and causality [14]. This passive surveillance system is managed by the Ministry of Health. As of 30 December 2022, a total of 38,757 AEFIs had been reported, with 97% classified as non-serious and 3.0% as serious [15]. These data are periodically reported by national health authorities in public bulletins and online platforms. Factors such as age, sex, vaccine type, and number of doses administrated may influence the likelihood of experiencing serious adverse effects [16,17,18]. Despite this national framework, state-level analyses remain limited. Nuevo León, a highly urbanized and densely populated state in northern Mexico, has played a central role in the vaccination rollout. However, the profile of AEFI in this population, including the frequency and factors associated with severity, has not been fully characterized. The availability of a regional pharmacovigilance database offers a unique opportunity to explore real-world AEFI patterns and identify possible risk factors in this context. Therefore, the objective of this study was to characterize the AEFIs associated with five COVID-19 vaccines (Pfizer/BioNTech, AstraZeneca, CanSino, Moderna, and Sinovac) using data from a pharmacovigilance system. Additionally, the study aimed to explore factors related to the type and seriousness of AEFIs reported in Nuevo León, Mexico.

## 2. Materials and Methods

This retrospective study was conducted using the AEFI surveillance database from the Mexican state of Nuevo León, located in northeastern Mexico. Nuevo León has a population of approximately 5.7 million inhabitants, with 86% residing in urban areas. This study was carried out at the State Ministry of Health, which is responsible for receiving and managing AEFI reports in accordance with national and international guidelines. All AEFIs, including UAEs, reported between December 2020 and June 2022 following administration of at least one dose of a COVID-19 vaccine were reviewed (n = 2213). The sample size provided a statistical power greater than 95% at a 95% confidence level.

AEFI reports were submitted using a standardized Case Report Form provided by the Mexican Ministry of Health. This form includes demographic data (age, sex), vaccine information (type of vaccine, number of doses administered), description of symptoms (local or systemic), date of vaccination, date of symptom onset, and seriousness of the event. For serious cases, the form must be accompanied by supporting medical documentation (e.g., clinical records, diagnostic test results) to facilitate case investigation and causality assessment. Reported symptoms were classified as either local (e.g., pain, redness, induration, warmth, itching, or axillary lymphadenopathy at the injection site) or systemic, based on organ system involvement. Symptoms included the following: generalized (chills, fever, headache, lethargy, fatigue, dizziness, irritability); digestive (poor appetite, nausea, vomiting, odynophagia, abdominal pain, diarrhea); respiratory (rhinorrhea, cough, dyspnea, bronchospasm); dermatological (skin reaction including exanthema, pruritus, edema and red or itchy eyes); cardiovascular (tachycardia, syncope); neurological (seizures, paralysis, Guillain–Barré syndrome); musculoskeletal (myalgia, joint pain, limited range of motion); and hemorrhagic (bleeding, thrombocytopenic purpura).

AEFI seriousness was classified according to WHO guidelines, which define a serious event as one resulting in death, life-threatening illness, hospitalization, or prolonged hospital stay; persistent or significant disability; or congenital anomaly [19]. Mexican regulations further stipulate that the event must occur within 30 days post-vaccination to be considered serious [14].

Causality assessments were performed using the WHO AEFI Causality Assessment Tool [10], which applies a standardized algorithm to determine the likelihood of a causal relationship between the vaccine and the adverse event. Cases were classified as consistent with vaccination, indeterminate, coincidental, or unclassifiable. Although initial reports are based on symptoms, serious AEFI cases typically undergo medical evaluation to establish a definitive diagnosis, which is then incorporated into the final classification; for example, a patient initially presenting with tachycardia and syncope may ultimately be diagnosed with myocarditis.

### Statistical Analysis

Descriptive statistics were used to determine the frequency distribution of AEFI by sex, age group, vaccine type, and number of doses received. When a single individual reported multiple adverse events, each symptom was analyzed independently. Binary multivariate logistic regression was employed to examine associations between the independent variables (sex, age, vaccine type, and number of doses) and specific types of AEFIs, such as arm pain or grouped systemic symptoms (e.g., digestive). Each AEFI category was analyzed as a dependent variable (coded 0 = yes, 1 = no). A separate regression model was used to assess the association of the same independent variables with the seriousness of the event (coded 0 = non-serious, 1 = serious). Odds ratios (ORs) and 95% confidence intervals (CIs) were reported for all models.

## 3. Results

A total of 2213 AEFI reports were registered in Nuevo León between December 2020 and December 2022. Of these, 97.3% were classified as non-serious and 2.7% as serious. The mean age of participants was 46.5 years (SE ± 18.6). The age distribution was as follows: ≤18 years, 2.1%; 19–39 years, 38.8%; 40–59 years, 26.9%; and ≥60 years, 32.2%. Among all AEFI cases, 79.1% were female and 20.9% male.

The majority of AEFI reports corresponded to the Pfizer/BioNTech vaccine (45.1%), followed by AstraZeneca (37.6%), Sinovac (8.9%), CanSino (5.6%), and Sputnik V (2.8%). This distribution is consistent with the local vaccination uptake, where Pfizer and AstraZeneca represented the most administered vaccines in the state during the study period. A single dose regimen was reported in 75.3% of participants.

The most frequently reported local AEFI was arm pain, while generalized symptoms were the most common systemic AEFIs. No deaths were reported among serious AEFI cases during the surveillance period. However, information regarding hospitalization was not consistently available in the dataset and could not be reliably analyzed.

Most serious AEFIs were reported within the first 72 h following vaccination. Although time-to-onset in days was not systematically recorded for all cases, this early timeframe was consistently observed in the available reports.

### 3.1. Associated Factors with AEFI Type

The association between sex and AEFI type was inconsistent. Female sex was associated with a higher likelihood of experiencing arm pain, digestive symptoms, and skin manifestations, but with a lower likelihood of neurological and hemorrhagic symptoms. Age ≥ 60 years was associated with reduced odds of presenting generalized, digestive, respiratory, cardiovascular, neurological, and musculoskeletal AEFIs. Receiving two or more vaccine doses increased the likelihood of experiencing arm pain, generalized symptoms, as well as digestive, respiratory, musculoskeletal, and hemorrhagic AEFIs (Table 1) (Appendix A).

### 3.2. AEFI Seriousness and Associated Factors

A total of 2.9% of AEFIs were classified as serious. Factors associated with an increased likelihood of serious AEFIs included age ≥ 65 years and receiving two or more vaccine doses. In contrast, women receiving the Pfizer/BioNTech vaccine were associated with a reduced likelihood of serious AEFIs (Table 2 and Table 3) (Appendix A).

Appendix A shows the results of the multiple linear regression model using the backward method with age, sex, and number of vaccine doses as independent variables, and severity of AEFI as the dependent variable. The results of the regression coefficients indicate a statistically significant model (Model 2), F = 7.319, *p* = 0.001, 95% CI [1.876–1.981] (Appendix A). Sex has a positive influence on the severity of AEFI symptoms, while the number of doses administered has a negative influence on the severity of AEFI symptoms.

## 4. Discussion

We analyzed pharmacovigilance data from Nuevo León to describe the association of sex, age, and vaccine type with the occurrence and seriousness of AEFI. Consistent with studies from United States [20,21,22,23], Brazil [17], Korea [18], and Argentina, AEFI reports were more frequent among women, possibly due to biological predispositions or reporting behaviors. Over 60% of reports involved individuals aged 19 to 59 years, aligning with Chen et al. [21], who noted stronger immune responses in younger adults. Reports from individuals ≤ 18 years were low (2%), consistent with frequencies reported elsewhere [16,17,21].

Vaccine distribution varied, with Pfizer/BioNTech (45%) and AstraZeneca (38%) predominating, reflecting availability and preferences seen internationally [16,17,22,23,24]. Arm pain and generalized symptoms such as headache, pyrexia, fatigue, and chills were the most common AEFIs, similar to prior findings [22,23,25]. Association between sex and specific AEFI types were inconsistent, while age ≥ 60 was linked to reduce odds of several systemic AEFIs, warranting further study of underlying mechanisms.

Vaccine type influenced adverse effect patterns: Pfizer/BioNTech increased arm pain risk but decreased hemorrhagic symptoms, whereas Sinovac and CanSino were associated with more musculoskeletal symptoms. Neurological AEFIs were rare (1.4%) and did not vary by vaccine type, contrasting with higher neurological event rates reported for Janssen vaccine recipients [24].

AEFIs were more common after the first dose; however, receiving two or more doses increased the odds of various symptoms and hemorrhagic events, consistent with Polack et al. [3] and Lee et al. Less than 3% of AEFIs were serious, similar to Brazil and Argentina (3.7%) [17], but lower than proportions reported in the United States and Canada [16,23]. Female sex was linked to a lower likelihood of serious AEFIs, in contrast to some studies [17,18], suggesting population or reporting differences.

Age ≥ 65 increased serious AEFI risk, in line with Brazilian data [17] and broader evidence on age-related vulnerability [18]. Pfizer/BioNTech vaccination was independently associated with reduced serious AEFI risk, consistent with Lee et al. [18], though contrasting with Canadian data [16]. Second doses tripled serious AEFI probability, possibly due to enhanced immune responses and pre-existing conditions, similar to Canadian reports [16].

A recent study from Baja California, Mexico, examining six different COVID-19 vaccines, offers complementary real-world data, confirming predominance of mild to moderate AEFIs and low serious event frequency [26]. Both studies noted higher AEFI reporting in females and younger adults, emphasizing consistent demographic trends across Mexican regions. Differences in vaccine availability and population characteristics highlight the need for localized pharmacovigilance to guide tailored public health strategies, strengthening national vaccine safety understanding and confidence.

Our study contributes valuable real-world evidence supporting COVID-19 vaccine safety in Mexico highlighting mostly mild to moderate AEFIs and low serious event frequency. These findings reinforce vaccination campaign confidence, critical for pandemic control. Associations with demographic and vaccine factors underline the importance of personalized communication and monitoring. Continued enhancement of pharmacovigilance systems, including integration of clinical data is essential for better safety signal detection and response.

In addition to demographic patterns, international studies have identified specific adverse reactions associated with COVID-19 vaccine products. For example, myocarditis has been reported after mRNA vaccines, particularly in young males [27,28]; thrombosis with thrombocytopenia syndrome (TTS) has been linked to ChAdOx1 nCoV-19 [29]; and Guillain–Barré syndrome (GBS) has been associated with Ad26.COV2.S [30]. In contrast, our analysis of state level reports from Nuevo León did not reveal unexpected safety signals beyond those already recognized in international literature, nor did they identify higher frequencies of these conditions.

Despite limitations of passive reporting and missing data on comorbidities and the inability to establish causal inference, our findings indicate that the local safety profile of COVID-19 vaccines in Nuevo León is consistent with global evidence. This work informs policymakers and healthcare providers optimizing vaccination and addressing hesitancy, while reinforcing the importance of contextualized pharmacovigilance to guide public health communication and maintain vaccine confidence.

## 5. Conclusions

In Nuevo León, most AEFIs following COVID-19 vaccination occurred in women and adults aged 19–59 years, predominantly after the first dose, and were generally mild and self-limiting. Serious AEFIs represented less than 3% of reports. Older adults and second vaccine doses were associated with higher odds of serious events, while female sex and Pfizer/BioNTech vaccination were associated with reduced odds. Overall, no unexpected safety signals beyond those already described in the international literature were identified. These results contribute to understanding the safety profile of COVID-19 vaccines at the state level in Mexico and provide evidence to support ongoing surveillance and local risk communication.

## Figures and Tables

**Table 1 idr-17-00108-t001:** AEFI type. State of Nuevo León AEFI Notification System, December 2020–June 2022 (n = 2213).

AEFI Type	n	%
Local		
Arm pain	871	39.4
Redness/very warm injection site	252	11.4
Induration	98	4.4
Itching at the injection site	62	2.8
Lymphadenopathy	40	1.8
Systemic		
Generalized	1737	78.5
Musculoskeletal	835	37.7
Digestive	756	34.2
Respiratory	247	11.2
Dermatological	174	7.9
Cardiovascular	134	6.1
Neurological	30	1.4
Hemorrhagic	13	0.6

**Table 2 idr-17-00108-t002:** Associated factors with AEFI type. State of Nuevo León AEFI Notification System, December 2020–June 2022 (n = 2213).

	Associated Factors with AEFI Type		
	OR Adjusted (95% CI)		
	Arm pain	Generalized	Digestive
Sex, female	1.30 (1.10, 1.70) **	1.0 (0.80, 1.30)	1.30 (1.00, 1.50) *
Age, ≥60	0.96 (0.86, 1.07)	0.78 (0.69, 0.89) ***	0.84 (0.75, 0.94) **
Vaccine type			
AstraZeneca	Ref.	Ref.	Ref.
CanSino	1.10 (0.70, 1.90)	0.61 (0.35, 1.06)	0.83 (0.52, 1.33)
Moderna	1.20 (0.80, 1.90)	1.10 (0.60, 2.00)	0.77 (0.50, 1.17)
Sinovac	0.53 (0.33, 0.85) **	0.43 (0.29, 0.64) ***	1.40 (0.90, 1.90)
Pfizer/BioNTech	1.80 (1.50, 2.20) ***	0.55 (0.43, 0.70) ***	0.83 (0.68, 1.02)
≥2 doses	1.40 (1.10, 1.70) **	1.30 (1.00, 1.70) *	1.50 (1.20, 1.90) ***
	Respiratory	Dermatological	Cardiovascular
Sex, female	1.10 (0.80, 1.50)	1.50 (1.03, 2.30) *	1.10 (0.70, 1.70)
Age, ≥60	0.81 (0.68, 0.96) *	1.10 (0.90, 1.30)	0.57 (0.45, 0.72) ***
Vaccine type			
AstraZeneca	Ref.	Ref.	Ref.
CanSino	0.32 (0.11, 0.90) *	1.50 (0.70, 3.30)	0.36 (0.11, 1.21)
Moderna	0.67 (0.35, 1.28)	1.20 (0.60, 2.70)	0.56 (0.23, 1.39)
Sinovac	1.80 (1.10, 2.9) **	1.70 (0.90, 3.10)	1.20 (0.50, 2.50)
Pfizer/BioNTech	0.84 (0.62, 1.14)	1.40 (1.00, 2.00)	1.00 (0.70, 1.60)
≥2 doses	1.40 (1.10, 2.00) *	1.30 (0.90, 1.90)	0.85 (0.56, 1.29)
	Neurological	Musculoskeletal	Hemorrhagic
Sex, female	0.31 (0.15, 0.65) **	1.00 (0.80, 1.30)	0.13 (0.04, 0.49) **
Age, ≥60	0.32 (0.18, 0.57) ***	0.86 (0.77, 0.96) **	1.00 (0.50, 2.10)
Vaccine type			
AstraZeneca	Ref.	Ref.	Ref.
CanSino	0.50 (0.060, 4.07)	1.70 (1.40, 2.00) ***	--
Moderna	0.72 (0.19, 2.76)	1.50 (0.90, 2.30)	0.72 (0.13, 3.96)
Sinovac	--	1.90 (1.30, 2.80) **	0.81 (0.10, 6.76)
Pfizer/BioNTech	0.72 (0.31, 1.66)	1.10 (0.80, 1.70)	0.20 (0.04, 0.98) *
≥2 doses	0.60 (0.25, 1.45)	2.00 (1.60, 2.40) ***	4.40 (1.30, 14.50) *

* *p* ≤ 0.05, ** *p* ≤ 0.01, *** *p* ≤ 0.001. OR = odds ratio.

**Table 3 idr-17-00108-t003:** Associated factors with AEFI seriousness. State of Nuevo León AEFI Notification System, December 2020–June 2022 (n = 2213).

	AEFI		OR Adjusted (95% CI)
	Serious	Non-Serious	
Sex, female	50.00%	74.30%	0.42 (0.26, 0.70) ***
Age, ≥65	28.10%	19.50%	2.30 (1.20, 4.20) ***
Vaccine type			
AstraZeneca	54.70%	38.80%	Ref.
CanSino	0.00%	0.00%	--
Moderna	18.80%	5.30%	1.80 (0.80, 3.70)
Sinovac	1.60%	7.00%	0.16 (0.02, 1.18)
Pfizer/BioNTech	25.00%	48.80%	0.37 (0.20, 0.70) ***
≥2 doses	51.60%	23.90%	3.30 (1.90, 5.60) ***

*** *p* ≤ 0.001. OR = odds ratio.

## Data Availability

The raw data supporting the conclusions of this article will be made available by the authors on request.

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
