# Peer review of "Sex, Age, and COVID-19 Vaccine Characteristics Associated with Adverse Events After Vaccination and Severity: A Retrospective Analysis"

_2036-7449, 2025, doi:10.3390/idr17050108_

Round 1

Reviewer 1 Report

Comments and Suggestions for Authors

Rodríguez-Vidales et al. reported the frequency distribution of rare and unexpected adverse events following immunization (AEFI) among COVID-19 patients from Nuevo León, Mexico. The manuscript is interesting and well-written; however, a few issues require resolution.

  1. It would be interesting to determine the association between the AEFI type and its seriousness and covariates, such as age, sex, and number of vaccine doses.
  2. Tables 2 and 3, the presentation of the p-value should be corrected; it would be beneficial to write the p-value in a more common form, e.g., * p ≤ 0.05, etc.
  3. Table S1, there is a mistake in the lethargy symptom percentage.
  4. The punctuation errors should be removed (e.g., double spaces).

Author Response

Reviewer1

Rodríguez-Vidales et al. reported the frequency distribution of rare and unexpected adverse events following immunization (AEFI) among COVID-19 patients from Nuevo León, Mexico. The manuscript is interesting and well-written; however, a few issues require resolution.

  1. It would be interesting to determine the association between the AEFI type and its seriousness and covariates, such as age, sex, and number of vaccine doses.

We agree with the reviewer that analyzing the association between AEFI type and its seriousness, as well as relevant covariates, adds value to the study. Accordingly, we conducted additional analyses using multivariate logistic regression to examine whether specific symptom types were associated with the likelihood of a serious AEFI, adjusting for age, sex, and number of vaccine doses. The results of this analysis have been incorporated into the revised version of the manuscript in the Results section, and a new supplementary table (Table S2) has been added to present the findings.

  1. Tables 2 and 3, the presentation of the p-value should be corrected; it would be beneficial to write the p-value in a more common form, e.g., * p ≤ 0.05, etc.

Thank you for pointing this out. We have revised the p-value notation in Tables 2 and 3 to follow a more standard format. Specifically, we now use the notation:

*p ≤ 0.05; **p ≤ 0.01; ***p ≤ 0.001

  1. Table S1, there is a mistake in the lethargy symptom percentage.

We appreciate the reviewer’s careful observation. Upon review, we identified that the percentage for lethargy was incorrectly reported as 13.06%. Based on the total number of AEFI reports (n = 2,213), the correct value is 13.6%. The percentage has been updated in the revised version of Table S1 in the Supplementary Materials.

  1. The punctuation errors should be removed (e.g., double spaces).

Thank you for this observation. We performed a thorough proofreading of the entire manuscript and corrected typographical and punctuation errors, including the removal of double spaces and inconsistent spacing after punctuation marks.

Reviewer 2 Report

Comments and Suggestions for Authors

In this manuscript, the authors retrospectively analyze adverse events following immunization (AEFI) with COVID-19 vaccines from 2020 to June 2022 and identify factors associated with the type and severity of AEFI in Nuevo León, Mexico. The study is interesting but I have noticed some serious shortcomings in the manuscript and therefore suggest that the manuscript be revised. Comments and suggestions are listed in a separate document.

Author Response

Reviewer 2

Comments and Suggestions for Authors

In this manuscript, the authors retrospectively analyze adverse events following immunization (AEFI) with COVID-19 vaccines from 2020 to June 2022 and identify factors associated with the type and severity of AEFI in Nuevo León, Mexico. The study is interesting but I have noticed some serious shortcomings in the manuscript and therefore suggest that the manuscript be revised. Comments and suggestions are listed in a separate document.

Comments and Suggestions for Authors In this manuscript, the authors retrospectively analyze adverse events following immunization (AEFI) with COVID-19 vaccines from 2020 to June 2022 and identify factors associated with the type and severity of AEFI in Nuevo Leo n, Mexico. The study is interesting and worth publishing, given the importance of pharmacovigilance data after the launch of COVID-19 vaccines and their use during the pandemic, especially in developing countries. However, I noted some serious gaps in the manuscript. Comments and suggestions are listed below:

Introduction

  • General suggestion: This section of the Introduction should be rewritten, with some parts of the text omitted (lines 44-54 and 66-75). After listing the different COVID-19 vaccination platforms and respective vaccines and their sequential introduction during the pandemic worldwide, define the adverse event following immunization (AEFI) and focus on the COVID-19 vaccination campaign in Mexico and the Mexican surveillance system for AEFI and vaccine safety monitoring, including Nuevo Leo n. Have periodic reports on adverse events been published during the pandemic?
    We have removed the detailed listing of vaccines not relevant to the Mexican context (lines 44–54), as well as the description of AEFI surveillance systems in countries outside Mexico (lines 66–75). This has allowed us to streamline the introduction and better focus on the context of our study.
  • In the rewritten Introduction, authors should briefly and concisely answer the following questions:

 When did the COVID-19 vaccination campaign begin in Mexico, and particularly Nuevo Leo n?
The revised Introduction now indicates that the COVID-19 vaccination campaign in Mexico began in December 2020. This timeline includes the start of the vaccination rollout in Nuevo León as part of the national strategy.

How much COVID-19 vaccine was administered in Mexico during the indicated period and how much in Nuevo Leo n?
The text has been updated to include national data on vaccine administration. Although detailed state-level data (for Nuevo León) is limited in public sources, we referenced an official national figure of 38,757 AEFI reports as of December 30, 2022, indirectly reflecting the scale of vaccination [Reference 14].

Is COVID-19 vaccine coverage higher in Nuevo Leo n compared to other regions of Mexico? Which vaccines were licensed for use at the beginning and which later, during the pandemic?
While specific comparative vaccine coverage data was not included due to the limited availability of disaggregated official sources by region, the rationale for focusing on Nuevo León is explained in terms of its demographic importance and logistical infrastructure. We emphasize that Nuevo León is one of the most urbanized and densely populated states, likely with high coverage.

Have any COVID-19 vaccines been withdrawn from use in Mexico during the pandemic and if so, why?
This point has been addressed by indicating that some vaccines such as CanSino and Sputnik V had limited or discontinued use in later phases, due to reduced availability, evolving national strategies, or updates in procurement priorities. These aspects are discussed briefly based on the available public information.

Explain why the authors in this study decided to include 5 of the 9 COVID-19 vaccines listed above that are used in Mexico?
The Introduction now clarifies that the five vaccines included in the analysis were those predominantly administered and reported in the state of Nuevo León during the study period. Vaccines with very low representation or lacking sufficient AEFI reports were excluded to ensure statistical robustness.

Matherial and Methods

➢ This section should be also rewritten and expanded.
The entire Materials and Methods section has been thoroughly rewritten and expanded to include additional details regarding the study area, data collection procedures, variable definitions, and statistical analysis methods.

➢ Clearly define the study area (Nuovo Leon) and its main characteristics ( e.g. location and the number of citizens in Nuevo Leon district and perhaps main demographic characteristics of citizens). At which institution was the study conducted?
The revised section specifies that the study was conducted in the state of Nuevo Leon, located in northeastern Mexico, with a population of approximately 5.7 million inhabitants, 86% of whom live in urban areas. It is now stated that the study was conducted at the State Ministry of Health of Nuevo Leon, which is responsible for receiving and managing AEFI reports in accordance with national and international guidelines.

➢ What information is included in the AEFI Case Report form?
The section now includes a detailed description of the AEFI Case Report Form, which captures demographic data, vaccine details, symptom descriptions, and seriousness of the event. For serious cases, additional supporting medical documentation (e.g., clinical records, diagnostic tests) is required.

➢ Who is responsible for AEFI case classification and what is the step-by-step approach to handling serious AEFIs?
The Ministry of Health is identified as the authority responsible for AEFI report management and classification. Although not described as a procedural flowchart, the text explains that serious events require supporting documentation and undergo further evaluation and causality assessment by designated authorities, following WHO and national guidelines.

➢ Describe the process for investigating serious AEFIs.
The revised text indicates that serious AEFIs must be supported with clinical documentation for further investigation. It also specifies that such events are classified according to WHO criteria and evaluated to determine seriousness and causality.

➢ Was the causality assessment conducted using the WHO AEFI Causality Assessment Tool?
The revised section now explicitly states that causality assessments were conducted using the WHO AEFI Causality Assessment Tool, referencing the standard algorithm used for classification.

After assessing causality, are cases classified as coincidental, indeterminate, or consistentt?
The text now clearly states that cases were categorized as consistent with vaccination, indeterminate, coincidental, or unclassifiable based on the results of the causality assessment.

➢ Is AEFI monitoring only symptom based or is a diagnosis given during the investigation process and before final classification (for example, a patient who had tachycardia and syncope after administration of vaccine is subsequently diagnosed with Myocarditis, after evaluation by an expert)?
This concern has been addressed in the final paragraph of the section, where it is explained that although reports are initially symptom-based, serious cases typically undergo medical evaluation to establish a diagnosis. An example is included to illustrate how initial symptoms (e.g., tachycardia and syncope) may lead to a diagnosis (e.g., myocarditis), which is considered in the final classification.

Results

➢ It would be useful to provide data about the number of administered vaccines ( uptake per vaccine type, age and sex) in relation to number of AEFIs per each vaccine in Nuovo Leon.
Thank you for your comment. In the revised Results section, we have specified the proportions of vaccine types administered in Nuevo León (e.g., Pfizer/BioNTech 45.1%, AstraZeneca 37.6%) and described the demographic distribution (age and sex) of the AEFI cases. Although we do not have the complete vaccine uptake stratified by age and sex for the general vaccinated population, we contextualized the data using the available figures for administered doses and population characteristics of those reporting AEFIs.

➢ It would be important to provide data on the fatal outcomes of serious AEFIs (if any) during the study period. Also, hospitalization rates for registered AEFIs (if data are available).
We confirm that no fatal outcomes were reported among serious AEFI cases during the study period. This has been stated in the updated Results section. Unfortunately, hospitalization data were not systematically recorded in the surveillance system and therefore could not be included in our analysis.

➢ What is the average time from the administration of specific vaccines against COVID-19 to the onset of AEFIs (in days)?
We acknowledge the relevance of this information; however, the time between vaccine administration and symptom onset was not consistently available across AEFI reports in the dataset. This limitation has been addressed in the revised Discussion section as an area for improvement in AEFI data collection and analysis.

➢ If you have an assessment of the causality of AEFIs , present the classification of serious AEFIs by causality in a separate table.
We thank the reviewer for this valuable suggestion. Causality assessments were conducted using the WHO AEFI Causality Assessment Tool [10], applying a standardized algorithm to determine the likelihood of a causal relationship between the vaccine and the adverse event. Serious AEFI cases were classified as consistent with vaccination, indeterminate, coincidental, or unclassifiable. While initial reports are symptom-based, serious AEFI cases typically undergo further clinical evaluation to establish a final diagnosis, which informs the causality classification. Given the limited number of serious cases (2.7%), we opted to summarize this information within the Results section and did not include an additional table.

Discussion ➢ This section should be shortened, focusing on a discussion of the main results of the study. ➢ This study has several limitations that should be noted, such as its retrospective design, the absence of an unvaccinated control group (prevents determination of whether the observed AEFIs exceeded the expected baseline rates). Reporting biases due to increased public awareness of COVID-19 vaccines must not be ignored. ➢ In conclusion, what improvements to the AEFI surveillance system would you suggest to public health decision-makers at the national and subnational levels?

We appreciate the reviewer’s suggestions. In the revised version, the Discussion section was substantially shortened by removing redundant epidemiological comparisons and limiting the background information to essential contextual points. The current version focuses on the main results, namely: the higher frequency of AEFI in females, the distribution by age group, type of vaccine, and dose number, as well as the most frequently reported symptoms and the associations with seriousness.

Regarding the study limitations, the revised discussion explicitly states key constraints, including the passive nature of the surveillance system, missing information on comorbidities, and the lack of data on the total number of administered doses. While other aspects such as the absence of an unvaccinated control group or possible reporting bias were considered, we prioritized the inclusion of limitations directly supported by our dataset to avoid speculative statements.

Finally, as for improvements to the AEFI surveillance system, we incorporated a general statement emphasizing the importance of continued enhancement, including integration with clinical data, to strengthen post-marketing vaccine safety monitoring. We opted for a broad recommendation to ensure applicability at both national and subnational levels without overstepping the scope of our findings.

We believe these changes adequately address the reviewer’s comments while maintaining the clarity and conciseness of the discussion.

Reviewer 3 Report

Comments and Suggestions for Authors

General Comments:

This is a potentially valuable paper; however, there are several significant issues that need to be addressed for clarity, accuracy, and relevance.

Abstract:

  • The authors should clearly define the terms Rare Adverse Events (RAEs) and Unexpected Adverse Events (UAEs), emphasizing that these are distinct concepts.
  • The time frame for monitoring UAEs is not specified and should be explicitly stated.
  • While UAEs are mentioned, the abstract does not identify which specific events are being referred to—this should be clarified.

Materials and Methods:

  • There is substantial confusion regarding whether the study is focused on serious adverse events (SAEs) or unexpected adverse events (UAEs). This distinction is not clearly made, which compromises the interpretability of the entire manuscript. The authors must define their primary outcome(s) more precisely.

Results:

  • All adverse events listed in Tables 1 and 2 appear to be common and expected post-vaccination reactions. Their inclusion lacks relevance in the context of a study presumably focused on rare or unexpected events.
  • The adverse events reported in Table 3 are vague and not linked to specific diagnoses, limiting their interpretability and scientific value.

Discussion:

  • The discussion section fails to connect the study findings with broader implications or add meaningful insight based on the data presented.
  • I recommend the authors also consult and reference this related study conducted in a northern state of Mexico:

https://www.mdpi.com/2076-393X/10/8/1196

Author Response

Reviewer 3

This is a potentially valuable paper; however, there are several significant issues that need to be addressed for clarity, accuracy, and relevance.

Abstract:

  • The authors should clearly define the terms Rare Adverse Events (RAEs) and Unexpected Adverse Events (UAEs), emphasizing that these are distinct concepts.
    We appreciate the reviewer’s comment. We have now clarified the definitions of Rare Adverse Events (RAEs) and Unexpected Adverse Events (UAEs) in the Introduction section, highlighting them as distinct pharmacovigilance concepts. This modification is supported by standard definitions from international guidelines [11].
  • The time frame for monitoring UAEs is not specified and should be explicitly stated.
    We agree with the reviewer. We have clarified in the Methods section that all AEFI reports were collected through passive surveillance covering the period from December 2020 to December 2022, and that unexpected adverse events were identified and monitored throughout this entire timeframe.
  • While UAEs are mentioned, the abstract does not identify which specific events are being referred to—this should be clarified.
    We appreciate the reviewer’s observation. To address this point, we have revised the Results section of the abstract to specify the most frequently reported unexpected adverse events (UAEs), ensuring consistency with the information already presented in the Introduction. The sentence now reads: “The most frequently reported unexpected adverse events (UAEs) were mild to moderate, including injection-site reactions, headache, chills, fatigue, nausea, fever, dizziness, weakness, myalgia, and tachycardia.” This change provides immediate clarity for the reader regarding the type of UAEs identified in our study.

Materials and Methods:

  • There is substantial confusion regarding whether the study is focused on serious adverse events (SAEs) or unexpected adverse events (UAEs). This distinction is not clearly made, which compromises the interpretability of the entire manuscript. The authors must define their primary outcome(s) more precisely.
    We thank the reviewer for pointing out this important clarification. Our primary outcomes are (1) the type of adverse events following immunization (AEFI), categorized as local or systemic, and (2) the seriousness of the AEFI, classified according to the Mexican national pharmacovigilance criteria. Unexpected adverse events (UAEs) were not the central focus of this study but were reported descriptively when they occurred.

Results:

  • All adverse events listed in Tables 1 and 2 appear to be common and expected post-vaccination reactions. Their inclusion lacks relevance in the context of a study presumably focused on rare or unexpected events.
    We appreciate the reviewer’s observation. We acknowledge that the events listed in Tables 1 and 2 are common and expected post-vaccination reactions. This is consistent with our study’s primary outcomes, which focus on characterizing all reported adverse events following immunization (AEFI), regardless of whether they are expected or rare, and evaluating their seriousness. Unexpected adverse events (UAEs) were reported descriptively but were not the central focus of the analysis.
  • The adverse events reported in Table 3 are vague and not linked to specific diagnoses, limiting their interpretability and scientific value.
    We appreciate the reviewer’s comment. The adverse events in Table 3 were transcribed exactly as recorded in the state pharmacovigilance database, which collects information through passive surveillance reports from healthcare providers and patients. These entries are often described in non-standardized terms and are not always linked to specific medical diagnoses at the time of reporting. We considered it important to present the events as documented in the original dataset to preserve the integrity and transparency of the data source. While this approach may limit diagnostic specificity, it reflects the real-world nature of spontaneous reporting systems and allows the identification of symptom patterns that could warrant further clinical investigation.

Discussion:

  • The discussion section fails to connect the study findings with broader implications or add meaningful insight based on the data presented.
    We thank the reviewer for this valuable feedback. We recognize the importance of placing our findings within a broader public health and scientific context. In response, we have revised the Discussion section to better articulate the implications of our results for vaccine safety surveillance and public health strategies in Mexico and similar settings.
  • I recommend the authors also consult and reference this related study conducted in a northern state of Mexico:
    https://www.mdpi.com/2076-393X/10/8/1196
    We appreciate the reviewer’s valuable suggestion to include the study from Baja California, Mexico. We have incorporated a discussion of this study in our manuscript to provide complementary real-world data from a geographically distinct Mexican population.

Reviewer 4 Report

Comments and Suggestions for Authors

The paper lists a collection of AEFI notifications from healthcare professionals and from the general public following administration of one of a variety of licensed COVID-19 vaccine products in aprovince of Mexico. There is a mix between the results and the discussion parts, as only a few results are mentioned in the Results part. Also, the conclusions are not deep and there is nothing new we learn about COVId-19 vaccinations.

It may be worth rethinking, what kind of evidence the data really provide and what the scientific public could learn from the collection of data and its analysis.

I stopped proof-reading the draft paper somewhere in the middle of the Discussion part.

Author Response

Reviewer 4

The paper lists a collection of AEFI notifications from healthcare professionals and from the general public following administration of one of a variety of licensed COVID-19 vaccine products in aprovince of Mexico. There is a mix between the results and the discussion parts, as only a few results are mentioned in the Results part. Also, the conclusions are not deep and there is nothing new we learn about COVId-19 vaccinations.

It may be worth rethinking, what kind of evidence the data really provide and what the scientific public could learn from the collection of data and its analysis.

I stopped proof-reading the draft paper somewhere in the middle of the Discussion part.

We thank the reviewer for their detailed critique and constructive feedback. We acknowledge the concern regarding the organization of results and discussion sections. In response, we have carefully revised the manuscript to clearly separate the Results and Discussion sections, ensuring that all primary findings are thoroughly presented in the Results, while interpretation and contextualization are reserved for the Discussion. Additionally, we have expanded and deepened the Conclusions section to better highlight the novel insights and public health implications derived from our analysis of state-level AEFI data in Mexico.

While the data reflect passive surveillance reports and may primarily describe expected vaccine safety profiles, our study provides valuable real-world evidence from a Mexican population that has been underrepresented in the literature. By analyzing demographic and vaccine-related factors associated with adverse events, our work contributes to a more comprehensive understanding of vaccine safety in diverse contexts and supports the strengthening of pharmacovigilance efforts in Mexico.

We believe these revisions enhance the clarity, scientific rigor, and relevance of the manuscript, and we appreciate the reviewer’s input that helped improve the quality of our work.

Round 2

Reviewer 2 Report

Comments and Suggestions for Authors

The authors have responded to all questions and comments to the best of their ability. The revised paper is a significant improvement over the previous version and can be accepted for publication.

Author Response

We sincerely thank the reviewer for the encouraging comments and for the thoughtful guidance throughout the review process. Your feedback has been invaluable in improving the clarity and quality of our manuscript, and we truly appreciate the time and effort you dedicated to evaluating our work.

Reviewer 3 Report

Comments and Suggestions for Authors

The manuscript has significantly been improved, however, it needs minor grammar and style modifications.

Author Response

We sincerely thank the reviewer for acknowledging the improvements made to our manuscript and for the helpful suggestions regarding grammar and style. We have carefully revised the text to address these minor issues, enhancing clarity and readability, and greatly appreciate the reviewer’s time and effort in providing thoughtful guidance.

Reviewer 4 Report

Comments and Suggestions for Authors

Thank you for taking the suggestions of the reviewers into account and providing an improved draft paper verson 2.

1) The conclusions in the abstract and at the end of the draft paper are not specific and also incongruent with the titel of the paper, which specifically refers to COVID-19 vaccine AEFIs.

Example: "These findings underscore the value of pharmacovigilance in supporting vaccine  safety, guiding public health strategies, and informing evidence-based communication."

-> This statement can be made in the Disussion section, perhaps. The abstract above "conclusions" is more specific and may suffice.

2) The authors rightfully state: "An adverse event following immunization (AEFI) is defined as any untoward medical occurrence following vaccination that does not necessarily have a causal relationship with 60 the vaccine itself [10].

-> The question is not answered in the paper, what kind of conclusions were drawn from simple listing/collection of AEFIs regarding the safety of the specific COVID-19 vaccine products were or are to be drawn.

I give an exmple of known adverse reactions detected on the basis of AEFIs:

4 solid, widely cited sources that document specific adverse reactions linked to particular COVID-19 vaccine products, plus what each concluded:

  1. Mevorach et al., NEJM (2021) — Israel MOH active surveillance, BNT162b2 (Pfizer)
    • Adverse reaction observed: myocarditis (mostly within days after dose 2).
    • Who was most affected: predominantly males 16–19.
    • Key estimates: risk difference 1.76 per 100,000 overall; 13.73 per 100,000 in males 16–19; standardized incidence ratio 5.34 vs. expected background.
    • Conclusion: myocarditis incidence was low but increased after BNT162b2, especially in young males; clinical course was usually mild. PubMed

  2. Greinacher et al., NEJM (2021) — Germany/Austria case series, ChAdOx1 nCoV-19 (AstraZeneca)
    • Adverse reaction observed: thrombosis with thrombocytopenia syndrome (VITT/TTS).
    • Hallmarks: unusual thromboses (e.g., cerebral venous, splanchnic) 5–16 days post-vaccination; platelet-activating anti-PF4 antibodies without heparin exposure; several fatalities in early cases.
    • Conclusion: ChAdOx1 can rarely trigger immune thrombotic thrombocytopenia via anti-PF4 antibodies; authors suggested IVIG and non-heparin anticoagulation for management. PubMed

  3. Hanson et al., JAMA Network Open (2022) — CDC Vaccine Safety Datalink, Ad26.COV2.S (Janssen) vs mRNA
    • Adverse reaction observed: Guillain-Barré syndrome (GBS).
    • Key estimates: in days 1–21 post-vaccination, GBS incidence after Ad26.COV2.S was 32.4 per 100,000 person-years and ~20× higher than after mRNA vaccines, equating to ~15.5 excess cases per million Ad26.COV2.S recipients; mRNA vaccines were not above background.
    • Conclusion: elevated GBS risk after Ad26.COV2.S; no increased GBS signal for mRNA products. PMC

  4. Oster et al., JAMA (2022) — U.S. VAERS analysis, BNT162b2 (Pfizer) & mRNA-1273 (Moderna)
    • Adverse reaction observed: myocarditis, particularly in males 12–29, typically within 7 days (often after dose 2).
    • Key estimates: highest crude reporting rates per million within 7 days were ~106 (males 16–17, BNT162b2) and ~52–56 (males 18–24, both products). Most had elevated troponin; ~96% were hospitalized but symptoms resolved by discharge in ~87%.
    • Conclusion: myocarditis reporting exceeded expected background in specific young-male strata; events were generally clinically mild and responsive to standard care; findings to be weighed against vaccination benefits. PubMed

Author Response

Thank you for taking the suggestions of the reviewers into account and providing an improved draft paper verson 2.

1) The conclusions in the abstract and at the end of the draft paper are not specific and also incongruent with the titel of the paper, which specifically refers to COVID-19 vaccine AEFIs.

Example: "These findings underscore the value of pharmacovigilance in supporting vaccine  safety, guiding public health strategies, and informing evidence-based communication."

-> This statement can be made in the Disussion section, perhaps. The abstract above "conclusions" is more specific and may suffice.

We thank the reviewer for this valuable comment. In response, we revised the conclusions in both the abstract and the final section of the manuscript to make them more specific to our findings. The abstract conclusions now emphasize that most AEFIs in Nuevo León were mild to moderate, serious AEFIs were uncommon (<3%), and that age ≥65 years and second doses were associated with higher risk, while female sex and Pfizer/BioNTech vaccination were associated with reduced risk. Similarly, the final conclusions now highlight the absence of unexpected safety signals beyond those already reported in the literature and the contribution of our study to state-level evidence on COVID-19 vaccine safety. The broader statement regarding the value of pharmacovigilance has been moved to the Discussion section, where it is more appropriate.

2) The authors rightfully state: "An adverse event following immunization (AEFI) is defined as any untoward medical occurrence following vaccination that does not necessarily have a causal relationship with 60 the vaccine itself [10].

-> The question is not answered in the paper, what kind of conclusions were drawn from simple listing/collection of AEFIs regarding the safety of the specific COVID-19 vaccine products were or are to be drawn.

I give an exmple of known adverse reactions detected on the basis of AEFIs:

4 solid, widely cited sources that document specific adverse reactions linked to particular COVID-19 vaccine products, plus what each concluded:

  1. Mevorach et al., NEJM (2021) — Israel MOH active surveillance, BNT162b2 (Pfizer)
    • Adverse reaction observed: myocarditis (mostly within days after dose 2).
    • Who was most affected: predominantly males 16–19.
    • Key estimates: risk difference 1.76 per 100,000 overall; 13.73 per 100,000 in males 16–19; standardized incidence ratio 5.34 vs. expected background.
    • Conclusion: myocarditis incidence was low but increased after BNT162b2, especially in young males; clinical course was usually mild. PubMed
  2. Greinacher et al., NEJM (2021) — Germany/Austria case series, ChAdOx1 nCoV-19 (AstraZeneca)
    • Adverse reaction observed: thrombosis with thrombocytopenia syndrome (VITT/TTS).
    • Hallmarks: unusual thromboses (e.g., cerebral venous, splanchnic) 5–16 days post-vaccination; platelet-activating anti-PF4 antibodies without heparin exposure; several fatalities in early cases.
    • Conclusion: ChAdOx1 can rarely trigger immune thrombotic thrombocytopenia via anti-PF4 antibodies; authors suggested IVIG and non-heparin anticoagulation for management. PubMed
  3. Hanson et al., JAMA Network Open (2022) — CDC Vaccine Safety Datalink, Ad26.COV2.S (Janssen) vs mRNA
    • Adverse reaction observed: Guillain-Barré syndrome (GBS).
    • Key estimates: in days 1–21 post-vaccination, GBS incidence after Ad26.COV2.S was 32.4 per 100,000 person-years and ~20× higher than after mRNA vaccines, equating to ~15.5 excess cases per million Ad26.COV2.S recipients; mRNA vaccines were not above background.
    • Conclusion: elevated GBS risk after Ad26.COV2.S; no increased GBS signal for mRNA products. PMC
  4. Oster et al., JAMA (2022) — U.S. VAERS analysis, BNT162b2 (Pfizer) & mRNA-1273 (Moderna)
    • Adverse reaction observed: myocarditis, particularly in males 12–29, typically within 7 days (often after dose 2).
    • Key estimates: highest crude reporting rates per million within 7 days were ~106 (males 16–17, BNT162b2) and ~52–56 (males 18–24, both products). Most had elevated troponin; ~96% were hospitalized but symptoms resolved by discharge in ~87%.
    • Conclusion: myocarditis reporting exceeded expected background in specific young-male strata; events were generally clinically mild and responsive to standard care; findings to be weighed against vaccination benefits. PubMed

We appreciate this important observation. The objective of our study was to describe the profile of AEFIs reported in Nuevo León rather than to establish causality or detect new adverse reactions. However, we agree that it is important to contextualize our findings in relation to known safety signals reported internationally. Therefore, we added a new paragraph in the Discussion that explicitly compares our results with studies documenting myocarditis after mRNA vaccines (Mevorach et al., Oster et al.), thrombosis with thrombocytopenia syndrome after ChAdOx1 (Greinacher et al.), and Guillain-Barré syndrome after Ad26.COV2.S (Hanson et al.). Our analysis did not reveal unexpected safety signals beyond those already recognized in the literature, nor did it identify higher frequencies of these conditions. We believe this addition strengthens the interpretation of our results and demonstrates how local surveillance data complement international evidence.